# Circular RNAs in Viral Infection and Antiviral Treatment

**DOI:** 10.3390/cells13232033

**Published:** 2024-12-09

**Authors:** Xiaocai Yin, Hongjun Li, Yan Zhou

**Affiliations:** Institute of Medical Biology, Chinese Academy of Medical Science and Peking Union Medical College, Yunnan Key Laboratory of Vaccine Research and Development on Severe Infectious Disease, Kunming 650118, China; yinxiaocai2023@163.com (X.Y.); lihongjun@imbcams.com.cn (H.L.)

**Keywords:** circRNAs, virus, miRNA, antiviral, disease, biomarker, vaccines

## Abstract

Circular RNAs (circRNAs) are a class of noncoding RNAs that lack the 5′-cap structure and the 3′ poly(A) tail. Their distinguishing feature is that the 3′ and 5′ ends are covalently linked to form a closed circular structure. CircRNAs have a longer half-life and stronger ribonuclease resistance compared with linear RNA. Viral infections lead to the production of circRNA molecules through the transcription and splicing mechanisms of host cells. circRNAs are produced from the transcription and splicing of the viral genome or from the splicing reactions of the host cell gene. They participate in regulating the replication of many viruses, including coronaviruses, human herpesviruses, human immunodeficiency virus, and cytomegalovirus. CircRNAs regulate the infection process by modulating circRNA expression in host cells and affect cellular biological processes. Some circRNAs have been proposed as diagnostic markers for viral infections. In this review, we discussed the properties of virus-derived circRNAs, the biological functions of diverse viruses-derived and host circRNAs during viral infections, and the critical role of circRNAs in the host’s antiviral immune defense. Extensive research on the applications of circRNAs can help us better understand gene regulatory networks and disease mechanisms.

## 1. Introduction

The potential of circRNAs as a gene delivery method and vaccine platform has sparked interest following the recent coronavirus outbreak and the expedited licensing of mRNA vaccines by the U.S. Food and Drug Administration. As a recent addition to the noncoding RNA family, circRNAs are endogenous non-coding RNA (ncRNA) that have enormous research potential. In 1976, Sanger et al. showed viroids, pathogenic and infectious agents, mainly affecting plants, to be noncoding circRNAs [1]. CircRNAs were once considered an incorrectly spliced product generated during the splicing process [2]. Subsequent studies have confirmed that circRNA molecules are extensively distributed in human cells and are involved in gene expression [3,4]. CircRNAs are characterized by their single-stranded, covalently closed-loop structure, lacking a 5′ cap and a 3′ polyadenylic acid (polyA) tail, and are produced via a special alternative splicing method known as backsplicing [5]. Recently, numerous studies have identified the multiple functions of circRNAs, including acting as a protein scaffold or microRNA (miRNA) sponge, being translated into polypeptides [6], exerting biological effects by binding to proteins [7], serving as messenger RNAs for translation [8], competing with linear RNA [9], and regulating maternal gene expression [7]. CircRNAs are excellent choices as biomarkers due to these properties. These properties can be summarized as sequence retention, tissue specificity, high stability, and high abundance [10]. Unlike linear RNA, circRNAs are single-stranded, covalently closed RNA molecules. The closed-loop structure gives it a longer half-life and stronger RNase R resistance [4,7]. Furthermore, the levels of many circRNAs can vary in diseased tissues, patient plasma, and exosomes [11,12,13]. They may, therefore, be suitable candidates as therapeutic targets or diagnostic molecular markers. Numerous investigations have demonstrated their distinct expression features and important biological roles in cancer [14,15,16,17], blood system diseases [18], neurological diseases [19], cardiovascular diseases [20,21], and autoimmune diseases [22].

Viruses are noncellular microorganisms that are small in size, simple in structure, contain only one kind of nucleic acid (DNA or RNA), and need living cells to replicate. Viruses are incomplete entities that obtain the required resources and energy to function entirely from the metabolic system of the host cell. During the evolution process, viruses and hosts have evolved multiple strategies to antagonize each other through interactions [23]. Viral circRNAs produced in virus-infected cells may have a bidirectional regulatory function, acting on both the host and the virus itself [24]. Recent studies have identified virus-encoded circRNAs from both DNA and RNA viral genomes. These circRNAs produced by viruses may serve as potential biomarkers of viral infections, and the specific circRNAs expressed by host cells may possibly play a role in this process. Moreover, circRNAs are involved in regulating antiviral immune responses [25], viral replication [26], and the pathogenesis of infectious diseases [27,28].

In this review, we provide an overview of virus-encoded circRNAs and circRNAs in cells, focusing on the categories and characteristics of viral circRNAs and virus-derived circRNAs, and discuss the potential functions of host-encoded circRNAs and the possible mechanisms of their biological generation. The development of circRNAs as therapeutics to treat viral infections, the potential applications of circRNAs in gene editing, and the use of circRNA vaccines for disease treatment and prevention are also introduced.

## 2. Circular RNA Biogenesis and Classification

To complete the splicing process, linear RNA primarily joins the front and rear exons from end to end via the GU/AG region in the intron. CircRNAs are a unique class of ncRNA molecules that form a closed loop through reverse splicing, in contrast to conventional linear RNA. CircRNAs do not contain a 5′ end cap and a 3′ end poly (A) tail. Through exon or intron cyclization, the 3′ and 5′ ends are joined to produce a completely circular shape. Due to their specific structure, exonucleases cannot degrade them, making circRNAs more stable and conserved than linear RNA. CircRNAs are extremely stable and resistant to degradation by nucleases owing to their circular shape. There are notable distinctions in the splicing techniques, benefits, and drawbacks between circRNAs and linear RNAs (Figure 1). A screening technique, CRISPR-RfxCas13d, pointed out by Chen Lingling and colleagues, is based on CRISPR-Cas13d and can rapidly identify functional circRNAs [29]. This technique can successfully separate circRNAs from linear mRNA by targeting the backsplicing site (BSJ) of circRNAs with gRNA.

The primary models for circRNA formation include: (1) “lariat-driven circularization” or “exon skipping”; (2) “intron-pairing-driven circularization” or “direct backsplicing”; (3) circular intron RNA (ciRNA) formation; (4) RBP-dependent cyclization; and (5) variable cyclization mode similar to alternative splicing [5,30]. Through these five methods, three major classes of circRNAs are primarily formed: eciRNAs (exonic circRNAs) represent the most prevalent type of circular RNAs, composed of one or more contiguous exons; ciRNAs (intrinsic circRNAs) are formed from a single intron and do not include exon sequences; ElciRNAs (Exon–Intron circRNAs) encompass one or more exons along with one or more introns [31] (Figure 1B). After splicing, ecircRNAs are transported from the nucleus to the cytoplasm, while EIciRNAs and ciRNAs remain in the nucleus [32]. The sources of viral circRNAs are diverse and can come from different types of viruses, including DNA viruses, positive-strand RNA viruses and negative-strand RNA viruses. Different viruses can produce circRNAs with varying lengths and sequence compositions. For instance, the average length of circRNAs derived from Kaposi’s sarcoma-associated herpesvirus (KSHV) is 377.7 nucleotides (nt), whereas for Human cytomegalovirus (HCMV) it is 372.0 nt, and for Epstein–Barr virus (EBV) it is 454.2 nt. However, the length of circRNAs in host cells typically ranges from 250 to 500 nucleotides [33]. Additionally, according to statistics, 50% (1963/3912) of circRNAs have been reported to have a length of 200–500 bp, which is compatible with the features of circRNAs. Remarkably, 8% (311 out of 3912) of circRNAs span nearly the entire length of viral genomes that are ≥10 kbp [34]. Bioinformatics statistical analysis found a preference for a particular genome strand in viral circRNAs, with positive-strand DNA creating the majority of circRNAs in Epstein–Barr virus (EBV) and negative-strand DNA producing the most in KSHV [35]. Bioinformatics analysis tools can be used to further analyze whether other virus-encoded circRNAs also have genome strand preferences. The abundance of viral circRNAs varies depending on the stage of infection and the cell type. While the quantity of certain viral circRNAs may be low during the early stages of infection and increase gradually as the infection worsens, the abundance of other viral circRNAs may be higher [36].

CircRNAs from the host typically originate from the host cell’s own genome. They are derived from the host cell’s precursor mRNA (pre-mRNA) through a specialized splicing process, independent of any pathogen. In contrast, viral circRNAs may originate from pathogens. These circRNAs, encoded by the viral genome, are likely produced during the viral infection of host cells and participate in viral replication, pathogenesis or evasion of the host’s immune response. The functions of two types of circRNAs have both similarities and differences.

## 3. Biological Functions of circRNAs During Viral Infections

### 3.1. Role as a Molecular Sponge for miRNA

Host circRNAs act as miRNA sponges during virus replication, interacting with intracellular miRNAs and regulating cellular gene expression. By attaching to miRNAs, they can inhibit miRNA activity, prevent miRNAs from binding to their target genes and alter host gene expression. This mechanism can selectively regulate specific host genes, thereby providing a favorable environment for viral infections (Figure 2). CircRNA/microRNA regulatory axes play a crucial role in the development of Hepatitis B Virus (HBV)-associated Hepatocellular Carcinoma (HCC). Scholar Rui Liao has thoroughly examined the biological functions of circRNA_101764, circRNA_100338, circ-ARL3, and circ-ATP5H in HBV-HCC and how they modulate the progression of HBV-associated HCC (Table 1) [37]. These interactions implicate not only individual genes but also entire signaling pathways. Modulating specific circRNAs or miRNAs could potentially suppress tumor growth and metastasis, offering new therapeutic targets and facilitating the development of small molecule drugs or gene therapies for HBV-HCC patients.

CircRNAs act as molecular sponges for miRNAs and play a significant role in virus-induced cancers, providing a new perspective for understanding the molecular mechanisms of cancer. EBV is one of the most important causes of nasopharyngeal cancer. Based on bioinformatics analysis, some researchers have screened out circCRIM1 (circCRIM1, derived from the human CRIM1 gene, is a host-encoded circular RNA located on chromosome 2p22.2), which is markedly upregulated in the metastasis of nasopharyngeal cancer. Using in vivo and in vitro investigations, it has been verified that circCRIM1 competitively binds to miR-422a via the ceRNA mechanism. Consequently, FOXQ1 inhibition by miR-422a is lessened, which encourages neural progenitor cell (NPC) metastasis and docetaxel resistance, as well as the development of a prognostic model with significant therapeutic use [38]. With advancements in detection technology, circCRIM1 may emerge as a vital biomarker for the early diagnosis of nasopharyngeal carcinoma, potentially fostering the development of novel drugs or therapeutic strategies.

CircRNAs have potential roles in virus-induced immune responses, particularly by acting as miRNA sponges to modulate the activity of immune cells and inflammatory reactions. Li et al. analyzed circRNAs and miRNAs in the brain tissues of mice infected with Japanese encephalitis virus (JEV) using whole-genome transcription Illumina sequencing and discovered that brain tissues infected with JEV had a differential expression of hsa_circ_0000220 (hsa_circ_0000220 is a circular RNA encoded by the host genome). Although whether circRNAs are involved in JEV-induced neuroinflammation is unknown, the circRNA functions as an miR-326-3p sponge to promote the expression of inflammatory cytokines [39]. In essence, circRNAs exploit host miRNAs to dampen the immune system, thereby facilitating their survival and dissemination within the host.

CircRNAs may play a role in inducing pyroptosis and affecting viral replication by modulating the function of miRNAs. High-throughput sequencing of circRNAs during EV-A71 infection has led to the identification of differentially expressed hsa_circ_0045431 (a host-encoded circular RNA), which functions as a sponge to bind hsa_miR_584. This finding suggests that EV-A71 infection induces pyroptosis by activating the hsa_circ_0045431/hsa_miR_584/NLRP3 regulatory axis, thereby further affecting viral replication and providing a potential biotherapeutic target for treatment [40]. Host circRNAs can also initiate pyroptosis to eliminate infected cells and pathogens, representing a self-preservation mechanism of the host. By modulating this regulatory axis to reduce pyroptosis and viral replication, we may be able to develop novel therapeutic strategies to control EV-A71 infection.

CircRNAs regulate signaling pathways by targeting miRNAs and influence viral proliferation in host cells. The downregulated circRNA hsa_circ_0007321, a host-encoded circular RNA derived from the human DIS3L2 gene, was identified using RNA sequencing. It regulates replication by targeting miR-492 and the downstream gene NFKBID, a negative regulator of nuclear factor-κB (NF-κB). This miR-492/NFKBID/NF-κB signaling pathway reveals that circRNAs regulate ZIKV replication and better inhibit the virus from spreading in host cells [41]. These findings hold significant importance for understanding the complexity of cellular regulatory networks and the mechanisms of viral infection.

Elucidating these mechanisms is of significant importance for understanding the molecular basis of viral infections, the progression of tumors, and the host immune response, providing potential targets for the development of novel antiviral therapeutic strategies and intervention measures.

### 3.2. Regulation of the Viral Infection Process via Protein Interactions

CircRNAs play an important regulatory role in viral infections by directly or indirectly interfering with protein function. CircRNAs and proteins work together to exert biological effects, forming complexes that depend on sequence and 3-dimensional structure [42]. They can also form a circRNA–protein A/B ternary (or the above) complex, directly or indirectly promoting interactions or dissociating interactions [7]. Viral-derived circRNAs can bind to important signal transduction pathway proteins, including transcription factors and viral nucleoproteins (NPs). This can modify intracellular signal transmission processes, altering the physiological state of cells and accelerating the spread of viral infections.

Following the identification of circRNAs associated with influenza A virus (IAV) infection, researchers found that circVAMP3, produced by host cells, serves as a decoy for the viral protein NP and nonstructural protein 1 (NS1). Mechanistically, circVAMP3 directly inhibits viral replication by reducing the interaction between NP and polymerase within the viral ribonucleoprotein complex. It also restores interferon-β activation by alleviating the inhibitory effect of NS1 on RIG I or TRIM25, thereby enhancing innate immunity to IAV infection [43]. CircRNAs may serve as potential targets for antiviral therapy by modulating molecular interactions within host cells to bolster innate immune defenses against viral infections.

When studying oncogenic viruses like KSHV, researchers found that circ_0001400, encoded by the host genome, suppresses viral gene expression and production by interacting with the splicing factor PNISR. It also promotes cell growth to maintain latent infection by inhibiting apoptosis. Additionally, circ_0001400 may enhance the host immune response by modulating the expression of immune-related genes [44]. Further research is needed to determine if circRNAs maintain viral latency in host cells by interacting with other proteins, if this mechanism is present in other viral infections, and if the stability of circRNAs contributes to viral latency.

This combination of circRNAs and proteins can regulate the stability, ubiquitination, and phosphorylation status of cell-cycle proteins and the related cell cycle-regulatory signaling pathways. Mo et al. found that novel circRNF13 (circRNF13 is encoded by the host genome) plays an important role in the development of nasopharyngeal carcinoma (NPC) through the circRNF13–SUMO2–GLUT1 axis. CircRNF13 activates SUMO2 protein by binding to the 3′ UTR of the small ubiquitin-related modified protein-2 gene. After attaching to SUMO2, circRNF13 interacts with the glucose transporter. SUMOylation and ubiquitination of GLUT1 promotes the degradation of glucose transporter 1. The glycolysis-regulated AMPK/mTOR pathway is inhibited, ultimately leading to the proliferation and metastasis of nasopharyngeal carcinoma [45]. CircRNAs have a potential role in regulating host cell metabolism, and some viruses may facilitate their replication and dissemination by impacting the glycolytic processes of host cells. We can explore the possibility of inhibiting viral replication and malignant tumor behavior by targeting these pathways.

### 3.3. Proteins Translation

With an increased understanding of circRNA functions, researchers have found that circRNAs can be translated into protein. The translation of circRNAs is independent of the m7G cap. It occurs through various mechanisms, including internal ribosomal entry sites (IRES)-mediated, m6A modification-mediated, rolling circle amplification (RCA) translation, and UTR-mediated pathways [46,47,48]. Additionally, a study developed a methodical strategy to optimize circRNA translation by focusing on the following five factors: internal ribosome entry sites, vector structure, 5′ and 3′ untranslated sections, and synthetic aptamer recruitment. The translation initiation technique boosts circRNA protein production hundreds of times and is used to quickly build and test the properties that influence protein generation from synthetic circRNAs [3].

Researchers have discovered a circular RNA, circE7, in HPV-16-positive cervical cancer cells. This circRNA is derived from the HPV-16 virus and is formed by the back splicing of E6 and E7 mRNAs. CircE7 has a circular structure, can be modified by m6A, and is involved in protein translation as it binds to polysomes. It contains sequences of the E7 oncogene and can produce the E7 oncoprotein, a key component in HPV-induced carcinogenesis [49]. This finding is that one of the essential components in HPV-induced cell carcinogenesis, E7 oncoprotein, can be produced by circE7 via translation [50]. We can investigate the role of circE7 in carcinogenesis, and researchers might assess whether circE7 could serve as a new biomarker for early detection and therapeutic response monitoring in cervical cancer within clinical samples.

## 4. CircRNAs and the Host Immune System

There has been long-term interest in foreign circRNA structures that initiate the immunological response in the body. Howard Y. Chang reported that purified circRNAs, compared to linear RNAs with identical sequences, induced a significantly more pronounced expression of multiple innate immune genes. These genes include retinoic acid-inducible gene I (RIG-I), melanoma differentiation-associated protein 5 (MDA5), 2′-5′-oligoadenylate synthetase 1 (OAS1), and protein kinase R (PKR), upon transfection into HeLa cells. This enhancement of the innate immune response also improves cellular resistance to viral infections, as shown by reduced infection rates in challenge assays [51]. Further studies revealed that the m6A modification distinguishes circRNAs from other types, preventing their immunogenicity and blocking their interaction with RIG-I and K63 polyubiquitin chains, which would otherwise trigger an immune response [52]. By identifying the triphosphorylated structure at the 5′ end of viral RNA, RIG-I can activate the downstream signaling pathways, cause the release of interferon and other inflammatory proteins, and inhibit virus replication. Interestingly, immune activation by 5′-triphosphate linear RNA is more potent than that induced by exogenous circRNAs in suppressing virus replication [53]. The structural differences between linear RNA and circRNA that affect their abilities to activate immune responses suggest that we must consider the architectural features of RNA when designing vaccines and therapeutic strategies.

Researchers have also discovered the degradation mechanism of circRNAs when cells are infected by viruses. CircRNAs form a 16–26 bp double-stranded RNA stem-loop structure, which binds to the natural immune factor double-stranded RNA-dependent protein kinase PKR and participates in antiviral immunity. In healthy cells, the stem-loop structure of circRNA molecules combines with PKR molecules implicated in antiviral activities, and their activity is regulated to avoid excessive activity from inducing an immune response. Activated RNase L quickly breaks down circRNAs resulting from viral infection or poly I:C stimulation in infected cells, which is essential for the release of PKR [54]. RNase L can act on circRNAs under the influence of viral stimulation, cleaving and degrading it. Although the production rate of circRNAs is low and insufficient to fill the degraded circRNAs, PKR is released and participates in the antiviral immunity of cells [55,56,57]. In viral infection-simulation experiments, reducing circRNA expression releases double-stranded RNA-binding nuclear factors NF90/NF110 from the nucleus into the cytoplasm. This release neutralizes viral mRNA, contributing to the antiviral immune response [58]. The dynamic balance of intracellular RNA molecules suggests that modulating the expression or stability of circRNAs through drugs or gene therapy may enhance cells’ antiviral capabilities. Lastly, Chen Lingling demonstrated that the degree of immunological reactivity induced by circRNAs synthesized using various cyclization techniques varies. Immune responses are elicited by extra sequences introduced by group I introns to generate circRNAs, of which circRNAs formed by ligases elicit the smallest responses [57]. These findings suggest that circRNAs can trigger innate immune responses, specifically the ability of the host to identify viral circRNAs and build an immune response against viral infections, and that its immunogenicity can be utilized to develop circRNA self-adjuvanted vaccines.

## 5. Application of circRNAs to Treat Viral Infections

Due to its characteristics, circRNAs derived from viruses and hosts play regulatory roles in viral infection, modulation of the innate immune response, and other biological processes of antiviral immunity. Therefore, they are increasingly being used as therapeutic agents, vaccines, and molecular medicines to treat viral infections (Figure 3). Virus-encoded circRNAs with HBV infection serve as valuable biomarkers and targets for the diagnosis, prevention, and treatment [59]. They are also harnessed in the production of antiviral vaccines, where their role in the main preparation process is highlighted. Furthermore, the mechanism of circRNAs knockout is achieved through the use of CRISPR-Cas9 technology [60], while the degradation and virus typing of virus-derived circRNAs are facilitated by CRISPR-Cas13, which has been optimized and upgraded specifically for targeting the BSJ site of circRNAs.

### 5.1. CircRNAs as a Biomarker of Viral Infections

CircRNAs generated by viruses control how host and viral genes are expressed. They also function as biomarkers and potential treatment targets for numerous viral infections. Circ_3205 (circ_3205 is encoded by the SARS-CoV-2 viral genome) can be used as a diagnostic marker of novel coronavirus pneumonia. Research indicates that circ_3205 is exclusively expressed in positive samples following SARS-CoV-2 infection of host cells and shows a positive correlation with S protein mRNA and viral load. Furthermore, acting as a sponge for hsa-miR-298, circ_3205 may facilitate the spread of COVID-19 by upregulating downstream KCNMB4 and PRKCE mRNA [61]. Given the potential of circRNAs as biomarkers in viral infections, further research is warranted to investigate the detection and quantification of Circ_3205 as a biomarker for SARS-CoV-2 infection. The ACE2 gene likely serves as a receptor through which SARS-CoV-2 enters host cells [62]. The majority of SRY transcripts are found as circRNAs, which can suppress promoter activity in ACE2. CircRNAs may modulate ACE2 expression by regulating the transcription of genes associated with SRY. Notably, in addition to ACE2, other molecules such as AXL may serve as potential alternative receptors for SARS-CoV-2, facilitating viral entry. AXL may be a novel potential receptor that SARS-CoV-2 uses to infiltrate host cells. CircRNAs may influence viral entry by modulating the AXL signaling pathway and targeting microRNAs [63]. Increasing evidence shows that exosomal circRNAs can be used as a diagnostic and prognostic biomarker of various types of virus-related tumors [64,65]. Scientists are investigating the potential uses of circRNAs in disease detection and pharmacological therapy by examining the functions of circRNAs in viral infection, and by attempting to elucidate the expression pattern and association of circRNAs with host genes (Table 2). Additionally, we face the challenge of requiring highly specific and sensitive detection methods for circRNAs, whose expression can vary due to individual differences, disease stages, and types. This necessitates consideration of these variables in clinical applications to ensure the biomarkers’ universality and accuracy. It is anticipated that circRNAs will play a significant role in precision medicine in the near future.

### 5.2. CircRNAs and Antiviral Vaccines

CircRNAs, with their unique circular structure and stability, are emerging as the next generation of RNA-based vaccine platforms after mRNA vaccines. The circular structure of circRNAs not only enhances the stability of vaccines but also participates in cellular biological processes through various mechanisms, such as producing functional proteins via IRES (Internal Ribosome Entry Sites) and ORFs (Open Reading Frames), exemplified by circZNF609 [86] and circSfl [87]. Researchers have developed various in vitro synthesis techniques for circRNAs, primarily enzymatic ligation and chemical synthesis, as well as efficient circularization methods using self-splicing introns and T4 RNA ligase [88].

CircRNA vaccines exhibit immunogenicity due to their potential as self-adjuvanted vaccines. For instance, circRNA-based SARS-CoV-2 vaccines significantly enhance the expression of interleukin-6 and monocyte chemoattractant protein-1, stimulating innate immune responses [89]. Immunization with engineered circRNAs encoding protein antigens in mice leads to innate activation of dendritic cells in lymphoid tissues, antigen-specific CD8 T cell responses, and antitumor effects [90]. The circRNA vaccine incorporating a trimeric RBD from the spike protein elicits robust neutralizing antibodies and T-cell responses. It sustainably produces a higher yield of antigens, resulting in a greater proportion of neutralizing antibodies. Additionally, the induction of Th1-skewed immune responses signifies the activation of the innate immune system [91]. The mLNP-circRNA vaccine induces a strong immune response and is specific in tissue targeting; the design strategy of the mannose-modified lipid nanoparticles (LNPs) circRNA vaccine shows promise [92]. These findings highlight the potential of circRNA vaccines in stimulating strong innate and T-cell responses in tissues.

Delivery systems are crucial for the success of circRNA vaccines, including LNPs, virus-like particles, exosomes and viral vectors. When delivering circRNA vaccines, a suitable delivery system can enable circRNAs to escape surveillance by the autoimmune system and help deliver circRNA vaccines to their target sites for expression. Among all mRNA delivery vectors, LNP-based delivery systems are the most efficient and dominant and are routinely used for the delivery of circRNA vaccines and drugs [91,93], including the COVID-19 mRNA vaccines mRNA-1273 and BNT162b2. The circular structure of circRNAs can enhance vaccine stability, with circRNAs as the antigen wrapped in biodegradable LNPs. The LNP-wrapped SARS-CoV-2 circRNA vaccine (receptor-binding domain [RBD] dimer) has high stability and can be used at 4 °C or even after 6 freeze–thaw cycles for up to 6 months [94], overcoming the strict storage and transportation conditions that are often a requirement for RNA vaccines.

The breakthroughs in the artificial in vitro synthesis of circRNAs provide the possibility for the development of next-generation RNA vaccines. We foresee circRNA vaccines as a delivery option and their potential applications in disease treatment and prevention; however, circRNA vaccines are still in their early stages. Thus, their optimization, delivery, and applications require further development and evaluation.

### 5.3. CircRNAs and Gene Editing

CircRNAs exhibit significant application potential in gene editing. CircRNAs can be used as a carrier for gene editing tools (e.g., the CRISPR-Cas9 system) to carry the guide sequence (e.g., sgRNA) of the editing system to target cells, resulting in precise genome editing. An innovative CRISPR/Cas13a-induced exponential amplification detection method can effectively and specifically detect target circRNAs even in the presence of a large amount of linear RNA interference. This method has a broad application prospect and is not only limited to circRNAs. It can be easily expanded to other RNA targets by changing the guide RNA, such as mRNA splice variants and RNA viruses [95]. CRISPR-Cas9 may knockout circRNAs by disrupting the intron or ALU element pairing [31]. High-throughput CRISPR-Cas systems are employed for the functional screening of circRNAs, effectively distinguishing them from their homologous linear RNAs. Scholars have successfully constructed a Zbtb1 gene-knockout EL4 cell line using CRISPR-Cas9 technology and screened for differentially expressed microRNAs and circRNAs in normal and Zbtb1-deficient EL4 cell lines, providing potential targets for drug development and the clinical treatment of T-cell lymphoma [96]. The complexity of the biogenesis mechanism of circRNAs makes it difficult to determine which intronic sequences to target. Compared with RNAi-based strategies that directly target backward splicing junctions, using the CRISPR-Cas9 system has limitations in targeting intron sequences involved in circRNA production [31]. However, utilizing the CRISPR-Cas13d tool and optimizing the design of single-guide RNA strategies targeting the back-splicing junction (BSJ) sites of circRNAs, CRISPR-Cas13 can efficiently and specifically knock down circRNAs without affecting their cognate linear RNAs, thereby significantly enhancing the specificity of circRNA silencing [97]. This greatly facilitates the establishment of circRNA animal models. Concurrently, we must ensure the safety and efficacy of CRISPR-Cas13 technology in humans.

Certain circRNAs can be used as targets for gene editing, regulating their expression and function, and can be used for the detection and typing of viruses. For example, CRISPR-Cas13a was used to develop methods to detect HSV [98] and HIV-1, and a point-of-care testing (POCT) method was established in combination with reverse transcriptase-assisted amplification, providing a key POCT tool for determining blood glucose levels and for HIV-1 management [99]. Additionally, CRISPR-based portable detection methods have been developed for monitoring HDV and HEV RNA and are especially suitable for areas with limited resources [100,101]. However, CRISPR/Cas13a technology has some limits in measuring HDV RNA. To improve this, we can combine it with digital PCR and other methods. Also, it is a good idea to create tests that can find HDV along with other viruses like HBV and HCV at the same time. This will make the testing more complete.

## 6. Conclusions

This review provides recent developments in the role of circRNAs in viral infections and antiviral strategies, highlighting their significance in virology. Despite advancements, challenges remain. These include the lack of standardized detection, difficulties in functional analysis, specific sample preparation needs, and obstacles in clinical implementation. These efforts will lay a crucial foundation for the development of novel antiviral strategies and treatment methods. We believe that research on circRNAs will pave the way for the development of innovative antiviral strategies and treatments.

## Figures and Tables

**Figure 1 cells-13-02033-f001:**
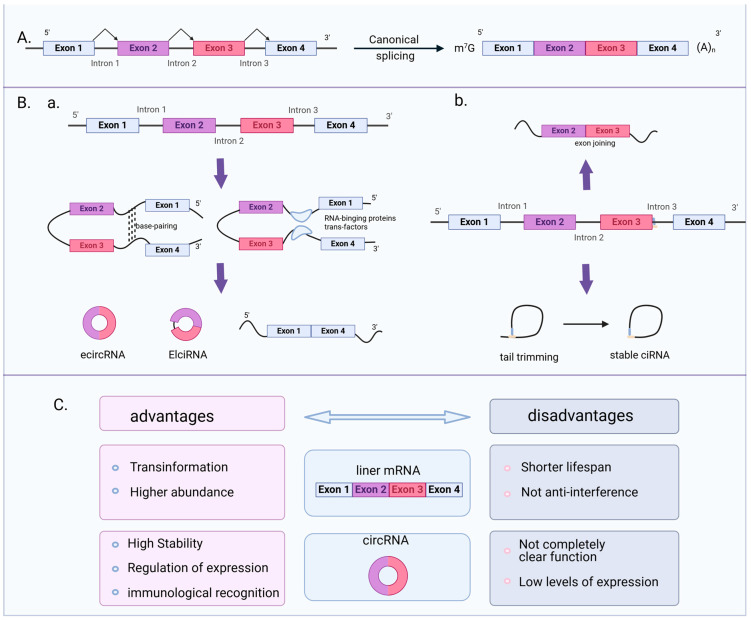
CircRNAs vs. Linear RNAs: splicing techniques, advantages, and disadvantages. (**A**). A schematic diagram of the traditional linear mRNA splicing process: The spliceosome cuts and joins the freshly generated precursor mRNA (pre-mRNA), eliminating introns and joining exons to create mature messenger RNA (mRNA). (**B**). A schematic diagram of the splicing formation of the three major classes of circRNAs: CircRNAs are single-stranded, closed circular RNA molecules produced by alternative splicing or exon/intron cyclization of mRNA precursors. (**a**). CircRNAs can form through intron pairing facilitated by RNA secondary structures or by certain RNA-binding proteins (RBPs) that promote back-splicing of specific exons, resulting in exonic circRNAs or Exon–Intron circRNAs. (**b**). During this process, introns form RNA secondary structures through complementary base pairing. The spliceosome recognizes and connects the 3′ and 5′ splice sites of these introns, resulting in a closed circular RNA molecule. (**C**). A comparative analysis of the advantages and disadvantages of traditional linear mRNA and circRNA. The image provides a comparative analysis of linear mRNA and circRNAs, outlining their advantages and disadvantages. Compared with linear mRNA, the main outstanding advantages of circular RNA are high stability, regulation of expression, and immunological recognition. However, circRNAs also face challenges, such as low expression levels and incompletely understood functions.

**Figure 2 cells-13-02033-f002:**
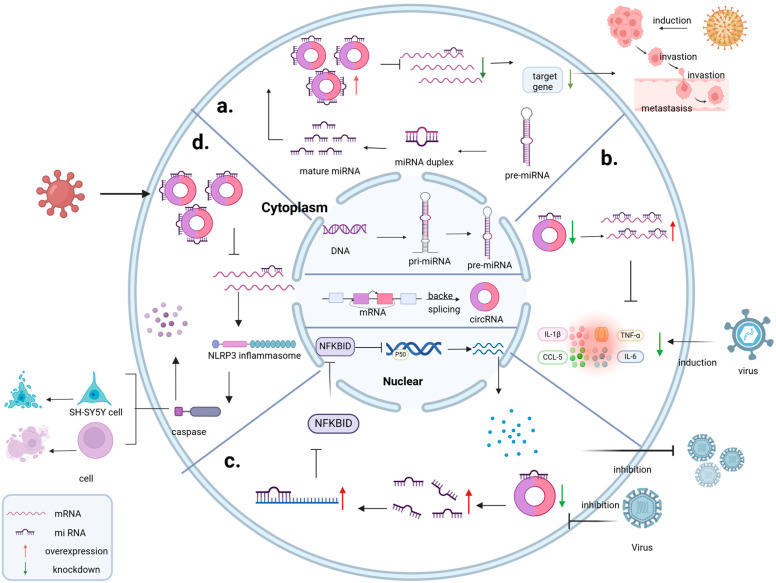
Regulation by competing endogenous RNAs (ceRNAs) in viruses. (**a**). ceRNAs influence the expression of host genes and regulate the virus-induced proliferation and invasion of tumor cells. (**b**). Influence of ceRNA regulation on immune cell activation and cytokine production to provide a more favorable environment for the virus. (**c**). circRNAs regulate signaling pathways by targeting miRNA and influence viral proliferation in host cells. (**d**). Host circRNAs eliminate infected cells and pathogens by triggering pyroptosis.

**Figure 3 cells-13-02033-f003:**
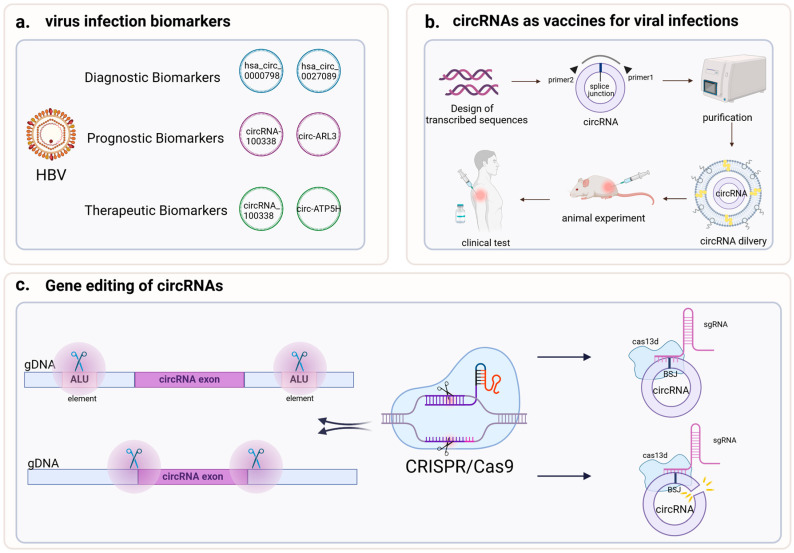
Main applications of circRNAs in antiviral therapy. (**a**). Virus-encoded circRNAs can be used as a biomarkers. (**b**). The use of circRNAs for antiviral vaccine production. (**c**). The related application of CRISPR-Cas 9 and CRISPR-Cas13 in circRNA.

**Table 1 cells-13-02033-t001:** Mechanisms and impacts of circRNAs in HBV-HCC.

CircRNA Name	Mechanism of Action	Impact on HBV-HCC	Reference
circRNA_101764	Interacts with miR-181	May promote the development of HBV-HCC	
circRNA_100338	Interacts with miR-141-3p	Closely correlated with metastatic progression of HBV-HCC	
circ-ARL3	Acts as a sponge for miR-1305	Promotes proliferation and invasion of HBV-HCC cells	[37]
circ-ATP5H	Sponges miR-138-5p	Promotes HBV-HCC development	
HBV_circ_1	Interacts with CDK1	Promotes proliferation, migration, inhibits apoptosis of HBV-HCC cells	

**Table 2 cells-13-02033-t002:** Roles, mechanisms, and effects of circRNAs in viral diseases.

Virus	CircRNA	Test Method	Different Diseases	Mechanism	Effect	Reference
human cytomegalovirus	hsa_circ_0001445 hsa_circ_0001206	unknown	Severe or even fatal diseases in newborns with immunodeficiency and adults with immunodeficiency	circRNA/miRNA/mRNA analysis	circRNA participates in the regulation of host cell secretion pathways, cell cycle, and apoptosis	[66]
Epstein–Barr virus	hsa_circ_0007637	qRT-PCR	Nasopharyngeal carcinoma	hsa_circ_0007637/miR-636/TPD52	High hsa_circ_0007637 expression predicted a poor outcome for NPC patients	[67]
influenza A virus	circVAMP3	Deep RNA sequencing	Influenza	Reduces the interaction between NP and polymerase alkalinity 1, polymerase alkalinity 2, or vRNA to interfere with the activity of viral ribonucleoprotein complexes	Directly inhibition of virus replication	[43]
influenza A virus	circMerTK	RNA sequencing	Influenza	circMerTK affects IFN-β signal generation and downstream signal transduction	Enhances replication of IAV	[25]
influenza Virus	circRNA AIVR	Reverse transcription-quantitative PCR	Influenza	circRNA AIVR/miR-330-3p/recombinant CREB binding protein	Increases IFN-β production and advances the understanding of the roles of circRNAs in the cellular innate antiviral response.	[68]
coxsackievirus	circ_0076631	Real-time quantitative PCR	Viral myocarditis and dilated cardiomyopathy	circ_0076631/miR-214-3p/cvb3	Suppression of viral translation	[69]
coxsackievirus group B5	novel_circ_0002006 novel_circ_0001066	Real-time quantitative PCR	Hand, foot and mouth disease	Two novel circRNA might act as a molecular sponge for miRNA through the IFN-I pathway and NF-κB pathway	Inhibition of CVB5 replication	[70]
Japanese encephalitis virus	circStrbp	RNA sequencing	Japanese encephalitis	circStrbp/miR709/CX3CR1	ceRNA pathway impacts JEV infection in vivo and in vitro	[27]
Japanese encephalitis virus	circ_0000220	Illumina RNA-sequencing	Japanese encephalitis virus-induced neuroinflammatory response	circ_0000220-miR-326-3p-BCL3/MK2/TRIM25	Knockdown of circ_0000220 or overexpression of miR-326-3p leads to a decrease in JEV-induced inflammatory cytokine production	[39]
enterovirus 71	hsa_circ_0045431	Quantitative real-time PCR	Hand foot and mouth disease	sa_circ_0045431/hsa_miR_584/NLRP3	Promotes inflammatory necrosis and virus replication	[40]
zika virus	hsa_circ_0007321	RNA sequencing	Congenital Zika syndrome	miR-492/NFKBID/NF-κB	Downregulates circRNA and inhibits NF-κB pathway. Promotes the replication of ZIKV.	[41]
human immunodeficiency virus	ciTRAN	RNA nanopore sequencing	Acquired immunodeficiency syndrome	HIV-1 hijacks ciTRAN to exclude serine/arginine-rich splicing factor 1 (SRSF1) from viral transcriptional complexes	Promotes effective viral transcription	[71]
hepatitis C virus	circular SERPINA3	Real-time qPCR	Hepatocellular carcinoma	circSERPINA3/miR-944/MDM2	Promotes the metastasis and oxidative stress of liver cancer cases	[72]
human adenovirus	hsa_circ_0002171	Unknown	Adenovirus pneumonia	circRNA mRNA regulatory network	hsa_circ_0002171 has significant value in diagnosing highly pathogenic pneumonia and severe highly pathogenic pneumonia	[73]
hepatitis B virus	hsa_circ_0028861	Microarray analysis	HBV-derived hepatocellular cancer	circRNA/miRNA/mRNA and downstream signaling pathway analysis	The combination of hsa_circ:0028861 and AFP shows better diagnostic ability	[74]
hepatitis B virus	circRNA_101764 circRNA_100338 circ-ARL3 circ-ATP5H	circRNA microarray and qRT-PCR	Hepatocellular carcinoma	circRNA_101764/miR-181/PI3K	Plays an important role in the cellular network during the development of HBV-HCC liver cancer	[37]
hepatitis B virus	hsa_circ_0027089	Quantitative reverse transcription polymerase chain reaction	Hepatitis B virus-related hepatocellular carcinoma	circRNA_100338/miR141-3p/MTSS1	Regulates the growth and metastasis of HCC cells	[75]
hepatitis B virus	circRNA1002	Quantitative reverse transcription PCR	Hepatocellular carcinoma	circ-ARL3/miR-1305/WNT2	circRNA1002 is involved in the progression of HCC, providing an improved early detection method for HCC	[76]
hepatitis B virus	hsa_circ_0003570 hsa_circ_0004018	Real-time quantitative polymerase chain reaction	Hepatitis B virus-associated hepatocellular carcinoma	circ-ATP5H/miR138-5p/TNFAIP3	Related to cell growth, tumor progression, invasiveness, and metastasis	[77]
hepatitis B virus	circ-ARL3	circRNA microarray	Hepatocellular carcinoma	hsa_circ_0027089/miR-15b-3p/OIP5	high circ-ARL3 was positively correlated with malignant clinical features and poor prognosis	[78]
hepatitis B virus	circRNA_10156	High-throughput RNA sequencing	Hepatitis B virus-related liver cancer	Enrichment of circRNA1002-related genes under GO conditions related to hormone pathways and cell–cell interactions	circRNA-10156 may be a promising therapeutic target for liver cancer treatment	[79]
human papilloma virus	circ0036602	qRT-PCR	Cervical cancer	circRNA/miRNA/mRNA	Promotes growth of CC cells	[80]
human papilloma virus	circE7	Reverse transcription-quantitative polymerase chain reaction	Cervical cancer	circ-ARL3/miR-1305/cancer gene	circE7 reduces E7 protein levels and inhibits cancer cell growth in vitro and in tumor xenografts	[50]
high-risk human papillomavirus	Circular RNA-mitochondrial tRNA translation optimization 1 (circMTO1)	Reverse transcription-quantitative polymerase chain reaction	Cervical cancer	Upregulation of miR-149-3p by consumption of ircRNA-10156, reduction in Akt1 expression, and inhibition of liver cancer cell proliferation	circMTO1 associated with clinical stage, tumor differentiation, lymph node metastasis, invasion depth, and independently linked with HR-HPV infection in CC.	[81]
Epstein–Barr virus	circBART2.2	Quantitative reverse transcription polymerase chain reaction	Nasopharyngeal carcinoma	Alters HMGB1 expression by sponging miR-34-5p and miR-431-5p	Crucial for regulating PD-L1 and subsequent immune escape in nasopharyngeal carcinoma	[82]
Epstein–Barr virus	ebv-circLMP2A	Reverse transcription-quantitative polymerase chain reaction and real-time PCR	EBV-associated gastric cancer	circE7 is localized in the cytoplasm through N6 methyladenosine (m6A) modification and translated to produce E7 oncoprotein	High expression of circRNA plays a crucial role in inducing and maintaining stemness phenotype, and is significantly associated with metastasis and poor prognosis	[83]
Epstein–Barr virus	circEAF2	Quantitative real-time PCR	Lymphoma	Serum miR-199a was downregulated in HR-HPV-positive CC patients and inversely correlated with circMTO1.	circEAF2 is a potential prognostic biomarker	[83]
Epstein–Barr virus	hsa_circ_0007637	High-throughput RNA sequencing (RNA-Seq)	Nasopharyngeal carcinoma	circBART2.2 activates transcription factors IRF3 and NF by binding to the helicase domain of RIG-I- κ B promoting the transcription of PD-L1	hsa_circ_0007637 expression distinguished NPC tissues from paired healthy tissues and NPC cell lines (HNE1 6-10B, 5-8F, CNE-2, and so on) from a normal epithelial (NP460) cell line.	[84]
merkel cell polyomavirus	circMCV-T	RNA enzyme R resistance RNA sequencing	Merkel cell carcinoma	ebv-circLMP2A/miR-3908/TRIM59/p53	Functional regulation of early transcriptional expression in regions important for virus replication and long-term persistence of the upper body	[85]

## Data Availability

No new data were created or analyzed in this study.

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
