# Peer review of "Circular RNAs in Viral Infection and Antiviral Treatment"

_cells, 2024, doi:10.3390/cells13232033_

Round 1
Reviewer 1 Report (Previous Reviewer 2)
Comments and Suggestions for Authors
The authors have improved the quality of the information in this new version, and have taken into account the recommendations that were assigned in the first round of review. Thus, the article is accepted for publication.
Reviewer 2 Report (Previous Reviewer 3)
Comments and Suggestions for Authors
The authors addressed all of my comments
and significantly improved the manuscript.
This manuscript is a resubmission of an earlier submission. The following is a list of the peer review reports and author responses from that submission.
Round 1
Reviewer 1 Report
Comments and Suggestions for Authors
The manuscript by Yin et al. is among the most problematic I have ever reviewed. It is evident from the beginning that the authors, who have no tracked record in this field, lack any solid understanding of virology or circRNAs. This is reflected in numerous inaccurate statements, use of nonscientific terminology, typographical errors, and nonsensical statements (see some examples below). In the abstract the authors state “In this review, we have discussed the properties of virus-derived circRNAs…,; however, the manuscript omits any references to several well-documented viral circRNAs. This is mainly caused by the authors' inability to comprehend the difference between viral circRNAs (encoded by the host genome) and host circRNAs that are dysregulated due to viral infection. Besides impressive figures, the paper has no scientific value. I strongly advise authors against publishing it in its current form, as this could have a significant negative impact on their reputation within the scientific community.
Line 14-15. ...” DNA viruses, including coronaviruses, human herpesviruses, human immunodeficiency virus, cytomegalovirus, and Kaposi’s sarcoma virus”
Coronaviruses and HIV are RNA viruses!!! The correct name is Kaposi’s sarcoma-associated herpesvirus and is a human herpesvirus, so is already included in “human herpesviruses”.
Line 240-241” Rolling circle translation requires lower amounts than messenger RNA…”
I don’t think the authors understand the term.
Line 248: … “multiribosomes”? I guess they mean polysomes.
Line 212….”circRNA preserves latent KSHV infection by blocking KSHV cleavage.”
What is KSHV cleavage?
Line 229: 3.3 “Messenger proteins and translation.” What are “messenger proteins”????
Line 231: Tanslation of circRNA does not depend… One example of numerous typos.
Line 297-299: “To better exploit the applications of circRNA, more researchers can more precisely investigate the reasons behind the significant disparities in immune responses caused by these circRNA generating methodologies.” This sentence does not make sense.
Line 419: “CircRNA is met with multiple challenges, in research related to viruses.” Wrong sentence structure.
Lines 423-424: “Despite these drawbacks, research on circRNA is ongoing worldwide.” What does it even mean?
Comments on the Quality of English LanguageThe language quality is poor leading to numerous confusing and incorrect statements.
Author Response
|
Response to Reviewer 1 Comments
|
||
|
1. Summary |
|
|
|
We would like to thank Cells for giving us the opportunity to revise our manuscript entitled “Circular RNA in viral infection and antiviral treatment”( ID number: cells-3180870). We thank the reviewers for their careful read and thoughtful comments on previous draft. Those comments are all valuable and very helpful for revising and improving our paper, as well as the important guiding significance to our researches. We have carefully taken their comments into consideration of preparing our revision and tried our best to improve the manuscript and made some changes in the manuscript. These changes will not influence the content and framework of the paper. The main corrections in the paper and the response to the reviewer’s comments are as flowing. |
||
|
|
|
|
|
2. Point-by-point response to Comments and Suggestions for Authors
|
||
|
Comments 1: Line 14-15. ...” DNA viruses, including coronaviruses, human herpesviruses, human immunodeficiency virus, cytomegalovirus, and Kaposi’s sarcoma virus” |
||
|
Response 1: Thank you for pointing this out. We agree with this comment. Therefore, We have removed "DNA" and "Kaposi’s sarcoma virus" from the sequence. On page one, line 15.
|
||
|
Comments 2: Line 240-241” Rolling circle translation requires lower amounts than messenger RNA…”I don’t think the authors understand the term. |
||
|
Response 2: Thank you for pointing this out. It is recognized that the objective of RCA is to increase the quantity of nucleic acid molecules, not to facilitate their translation. Utilizing RCA, researchers can more easily detect low-abundance circRNAs, which is beneficial for the investigation of circRNAs. We have revised the term "translate circRNAs" to "enhance the abundance of circRNAs". On page seven, line293.
Comments 3: Line 248: … “multiribosomes”? I guess they mean polysomes. Response 3: Thank you for pointing this out. We agree with this comment. Therefore, We have removed " multiribosomes " and " polysomes " from the sequence. On page seven, line302.
Comments 4: Line 212….”circRNA preserves latent KSHV infection by blocking KSHV cleavage.”What is KSHV cleavage? Response 4: Thank you for pointing this out. We agree with this comment. Therefore, We have removed " cleavage " and " genome splicing " from the sequence. Our statement is that circRNA serves a protective function by preventing the splicing of viral genes into functional mRNA, impeding the transcription of viral genes, and halting the production of new viral particles, which assists in maintaining the virus in a latent state. On page seven, line265.
Comments 5: Line 229: 3.3 “Messenger proteins and translation.” What are “messenger proteins”???? Response 5: Thank you for pointing this out. We agree with this comment. Therefore, The term "messenger proteins" may be misleading. We have corrected the term "messenger proteins" to "Protein Coding" to accurately convey that circRNAs can encode proteins and be translated into them. On page seven, line282.
Comments 6: Line 231: Tanslation of circRNA does not depend… One example of numerous typos. Response 6: Thank you for pointing this out. We agree with this comment. Therefore, We have removed " Tanslation " and " The tanslation " from the sequence. On page seven, line284.
Comments 7: Line 419: “CircRNA is met with multiple challenges, in research related to viruses.” Wrong sentence structure. Response 7: Thank you for pointing this out. We agree with this comment. Therefore, We have rephrased it to " Research on circRNAs face multiple challenges, particularly in the context of viruses ". On page eighteen, line491.
Comments 8: Lines 423-424: “Despite these drawbacks, research on circRNA is ongoing worldwide.” What does it even mean? Response 8: Thank you for pointing this out. We agree with this comment. Therefore, We have rephrased it to " Despite many uncertain challenges, research on circRNA is still ongoing worldwide.". On page eighteen, line497.
|
||
|
3. Response to Comments on the Quality of English Language |
||
|
Point 1: The language quality is poor leading to numerous confusing and incorrect statements. |
||
|
Response 1: Thank you for taking the time to provide feedback. We greatly value your input and understand that poor language quality can be frustrating. We are continuously working to enhance our language processing capabilities to deliver more accurate and clear information. |
||
Reviewer 2 Report
Comments and Suggestions for Authors
The work of Yin et al. makes an exciting review concerning the role of circular RNAs during viral infection events; in this work, the researchers expose the recent evidence that this type of nucleic acid regulates the replication processes of some viruses. Additionally, they show the implementation of this type of RNA for diagnosing or treating viral infections. In addition, they show how these RNAs can regulate the host's antiviral immune response. The review deals with a novel and interesting topic for the journal readers. The review is well structured and allows readers to follow the central theme adequately. However, there are some recommendations for publication in the journal.
• The cellular generation or synthesis of these types of RNA is unclear, and there is no adequate depth of this biogenesis process. It is recommended that they explain this idea better and that their scheme be more precise, emphasizing the steps for generating this type of RNA. It is suggested that they review this publication to get some ideas on this point. PMID: 37882652.
• It is recommended that a diagram of the therapeutic applications that cirRNA can have in viral infections be made; this will help to define these processes visually.
• In Figure 1, the figure caption is not delimited. It is recommended that each figure caption give a clearer explanation of the figure. In some cases, only the processes are mentioned in a general way, and they are not clear to the reader.
• Figure 2 is too complex. It is difficult to follow the route of where each process begins and stops. It is recommended that it be re-edited and that the key events that can contribute to the knowledge of cirRNA and viruses be placed.
• Finally, some ideas should be reviewed in detail. For example, in line 14 of the abstract, there is the following sentence: They participate in regulating the replication of many DNA viruses, including coronaviruses, HIV-1, etc. These viruses are not DNA viruses. These points should be corrected so as not to confuse the reader.
Author Response
|
Response to Reviewer 2 Comments
|
||
|
1. Summary |
|
|
|
We would like to thank Cells for giving us the opportunity to revise our manuscript entitled “Circular RNA in viral infection and antiviral treatment”( ID number: cells-3180870). We thank the reviewers for their careful read and thoughtful comments on previous draft. Those comments are all valuable and very helpful for revising and improving our paper, as well as the important guiding significance to our researches. We have carefully taken their comments into consideration of preparing our revision and tried our best to improve the manuscript and made some changes in the manuscript. These changes will not influence the content and framework of the paper. The main corrections in the paper and the response to the reviewer’s comments are as flowing.
|
||
|
2. Point-by-point response to Comments and Suggestions for Authors
|
||
|
Comments 1: The cellular generation or synthesis of these types of RNA is unclear, and there is no adequate depth of this biogenesis process. It is recommended that they explain this idea better and that their scheme be more precise, emphasizing the steps for generating this type of RNA. It is suggested that they review this publication to get some ideas on this point. PMID: 37882652. |
||
|
Response 1: Thank you for pointing this out. We agree with this comment. Therefore, We have delineated five modes of circRNA formation, briefly introduced the distinctions among the three major classes of circRNAs (eciRNAs, ciRNAs, and ElciRNAs), and expanded Figure 1-B. to describe the formation mechanisms of these three classes of circRNAs. On page two, lines 90 to 94.
|
||
|
Comments 2: It is recommended that a diagram of the therapeutic applications that cirRNA can have in viral infections be made; this will help to define these processes visually. |
||
|
Response 2: Thank you for pointing this out. The therapeutic applications of circRNA in viral infections represent an extensive area of research. In the fifth section, titled "Application of circRNA to Treat Viral Infections," the focus is on three major therapeutic applications: "CircRNA as a Biomarker of Viral Infections," "CircRNA and Antiviral Vaccines," and "CircRNA and Gene Editing," each accompanied by relevant illustrations for clarification. Furthermore, in the third section, "Biological Functions of circRNA During Viral Infections," numerous examples of potential therapeutic applications of circRNA in viral infections are presented, complete with corresponding figures to elucidate the content.
Comments 3: In Figure 1, the figure caption is not delimited. It is recommended that each figure caption give a clearer explanation of the figure. In some cases, only the processes are mentioned in a general way, and they are not clear to the reader. Response 3: Thank you for pointing this out. We agree with this comment. We have redrawn Figure 1, with each section separated and individually titled. From line 122 on page three to line 134 on page four.
Comments 4: Figure 2 is too complex. It is difficult to follow the route of where each process begins and stops. It is recommended that it be re-edited and that the key events that can contribute to the knowledge of cirRNA and viruses be placed Response 4: Thank you for pointing this out. We agree with this comment. Therefore, We have redrafted Figure 2, systematically and logically illustrating the role of circRNA as a molecular sponge for miRNA in the context of viral infections. Figure 2 is on page six.
|
||
Reviewer 3 Report
Comments and Suggestions for Authors
see attached file

NA
Author Response
|
Response to Reviewer 3 Comments
|
||
|
1. Summary |
|
|
|
We would like to thank Cells for giving us the opportunity to revise our manuscript entitled “Circular RNA in viral infection and antiviral treatment”( ID number: cells-3180870). We thank the reviewers for their careful read and thoughtful comments on previous draft. Those comments are all valuable and very helpful for revising and improving our paper, as well as the important guiding significance to our researches. We have carefully taken their comments into consideration of preparing our revision and tried our best to improve the manuscript and made some changes in the manuscript. These changes will not influence the content and framework of the paper. The main corrections in the paper and the response to the reviewer’s comments are as flowing.
|
||
|
2. Point-by-point response to Comments and Suggestions for Authors
|
||
|
Comments 1: In the abstract, specified viruses are not DNA viruses |
||
|
Response 1: Thank you for pointing this out. We agree with this comment. Therefore, We have removed "DNA" from the sequence.
|
||
|
Comments 2: Define EIciRNA ceRNAs c and others – these should appear when first mentioned. |
||
|
Response 2: Thank you for pointing this out. We agree with this comment. Therefore, We have provided full spellings or explanations for the first occurrence of terms such as EIciRNA, eciRNAs, and ceRNAs.
Comments 3: Expand on models to generate ciRNA - an illustration may be beneficial, as well as. examples for each model. Response 3: Thank you for pointing this out. We agree with this comment. Therefore, We have delineated five modes of circRNA formation, briefly introduced the distinctions among the three major classes of circRNAs (eciRNAs, ciRNAs, and ElciRNAs), and expanded Figure 1-B to describe the formation mechanisms of these three classes of circRNAs.
Comments 4: It would be beneficial to divide the review for host and viral ciRNAs; right now these are mixed and Response 4: We have annotated the known sources and definitions of circRNAs.
Comments 5: Elaborate of ciRNAs from viruses and their mode of functions Response 5: Thank you for pointing this out. This review encompasses circRNAs derived from both host genomes and viral origins. However, given the limited reports on viral-derived circRNAs, we have endeavored to highlight the known characteristics of viral-derived circRNAs in the second section titled "Circular RNA Biogenesis and Classification." Some examples of viral-derived circRNAs, such as circE7 and circ_3205, have been included in the article for illustrative purposes.
Comments 6: Not clear why some cRNAs are mentioned more in details while other are only mentioned within the table and others are skipped. Response 6: Thank you for pointing this out. The circRNAs selected for this review are highly relevant to the chosen themes or subheadings, and thus are elaborated upon as examples. The circRNAs listed in the table are primarily related to disease detection and therapy, and they serve to substantiate the notion of "CircRNA as a biomarker of viral infections."
Comments 7: Elaborate on the function of proteins derived from ciRNAs and their effects. In general, the vaccine section is not represented sufficiently and the authors move to quickly to delivery modes which may be a separate issue Response 7: Thank you for pointing this out. We agree with this comment. Therefore, We have reorganized this paragraph, The text begins by introducing the characteristics of circRNA and the synthetic technology of circRNA vaccines, discusses the immunogenicity of circRNA vaccines, including how they stimulate immune signaling and prevent viral infections, as well as the immune response and long-term effects of circRNA vaccines. It explains the delivery systems for circRNA vaccines and the stability of circRNA vaccines, and finally points out the future development prospects of circRNA vaccines.
Comments 8: If a report is cited and a ciRNA is mentioned, please elaborate more. For example, “Purified circRNA generated in vitro was transfected into mammalian cells, which effectively induced innate immunity genes and shielded the host from viral infections” Response 8: Thank you for pointing this out. We agree with this comment. Therefore, We have provided a detailed explanation for this citation.
Comments 9: In Fig 3 – pls provide examples ifor each application Response 9: Thank you for pointing this out. Sections "5.1 CircRNA as a Biomarker of Viral Infections," "5.2 CircRNA and Antiviral Vaccines," and "5.3 CircRNA and Gene Editing" all exemplify and substantiate their arguments around Figure 3. |
||
Reviewer 4 Report
Comments and Suggestions for Authors
This manuscript provides a comprehensive overview of current evidence on circRNAs. However, data presentation is not of easy reading in particular in the paragraph 3.Biological functions of circRNA during viral infection.
Although the three sub-paragraphs are helpful I would suggest to distinguish between well validated/useful, and weak/less impactful evidence.
The paragraph on viral infections is better organised due to the addition of Table 1. However, on this topic also, a comment by the Authors would add to the paper. Differentiating this from other reviews on the same topic may increase the relevance of the Authors effort.
Just an example: Page 17, line405. When discussing the CRISP/Cas13-assisted accurate and portable hepatitis D virus detection, I would have had a comment on the advantage of a rapid virus detection despite a possible sub-optimal sensitivity and on the need to validation in patients with HBV infection of genotypes different from "Asian" genotypes.
Page 2: lines 92-95. The second sentence is not clear. Are the findings based on statistic corroborating the results of the studies performed in huma, mice, macaques etc in vivo?
Finally, the role of circRNA in HBV infection needs a more accurate analysis, maybe a separate Table with a comment on the purpose of the use of one or another circRNA. PLease, refer to HBV section in the article: Current molecular biology and therapeutic strategy status and prospects for circRNAs in HBV-associated Hepatocelluar Carcinoma. Frontiers in Oncology 2021;11:697747.
Comments on the Quality of English Language
The quality is good, maybe minor editing may be required
Author Response
|
Response to Reviewer 4 Comments
|
||
|
1. Summary |
|
|
|
We would like to thank Cells for giving us the opportunity to revise our manuscript entitled “Circular RNA in viral infection and antiviral treatment”( ID number: cells-3180870). We thank the reviewers for their careful read and thoughtful comments on previous draft. Those comments are all valuable and very helpful for revising and improving our paper, as well as the important guiding significance to our researches. We have carefully taken their comments into consideration of preparing our revision and tried our best to improve the manuscript and made some changes in the manuscript. These changes will not influence the content and framework of the paper. The main corrections in the paper and the response to the reviewer’s comments are as flowing.
|
||
|
2. Point-by-point response to Comments and Suggestions for Authors
|
||
|
Comments 1:Although the three sub-paragraphs are helpful I would suggest to distinguish between well validated/useful, and weak/less impactful evidence. Response 1: Thank you for pointing this out. We regret to state that the extent of validation or impact size is difficult to define precisely. The circRNAs we have chosen to illustrate in detail are those that align closely with the selected themes or subheadings.
Comments 2: Just an example: Page 17, line405. When discussing the CRISP/Cas13-assisted accurate and portable hepatitis D virus detection, I would have had a comment on the advantage of a rapid virus detection despite a possible sub-optimal sensitivity and on the need to validation in patients with HBV infection of genotypes different from "Asian" genotypes. Response 2: Thank you for pointing this out. We have included our insights and comments on the cited references. “However, given the limitations of CRISPR/Cas13a technology in quantitative analysis, it is conceivable to explore the integration with digital PCR and other techniques to enhance the quantitative detection capacity of HDV RNA. Additionally, the development of multiplex detection methods capable of concurrently identifying HDV and other relevant pathogens, such as HBV and HCV, should be considered to bolster the comprehensiveness of the detection.”
Comments 3: Page 2: lines 92-95. The second sentence is not clear. Are the findings based on statistic corroborating the results of the studies performed in huma, mice, macaques etc in vivo? Response3: Thank you for pointing this out. This statement is a quotation from the article "Functions of Circular RNA in Human Diseases and Illnesses," specifically from the fourth line of the section 5.2 Length Distribution, Exon Numbers, and Strand Preference.
Comments 4: Page 2: lines 92-95. The second sentence is not clear. Are the findings based on statistic corroborating the results of the studies performed in huma, mice, macaques etc in vivo? Response4: Thank you for pointing this out. This statement is a quotation from the article "Functions of Circular RNA in Human Diseases and Illnesses," specifically from the fourth line of the section 5.2 Length Distribution, Exon Numbers, and Strand Preference. |
||
|
|
||
|
Comments 5: Finally, the role of circRNA in HBV infection needs a more accurate analysis, maybe a separate Table with a comment on the purpose of the use of one or another circRNA. |
||
|
Response5: Thank you for pointing this out. We also deeply agree on the significant role of circular RNA in HBV infection and have carefully studied the literature you recommended. We have incorporated our insights into the mechanisms of circular RNA in HBV infection into the section 3.1 Role as a Molecular Sponge for miRNA.
|
||
|
3. Response to Comments on the Quality of English Language |
||
|
Point 1: The quality is good, maybe minor editing may be required |
||
|
Response 1:Thank you for your kind words. I'm glad you found the quality satisfactory. If there are any areas that could use a bit of polishing, please don't hesitate to let me know. I'm always eager to improve my work. |
||
Round 2
Reviewer 1 Report
Comments and Suggestions for Authors
None
Comments on the Quality of English LanguageNone
Reviewer 4 Report
Comments and Suggestions for Authors
Thank you for your reply.